# Molecular Mechanisms of Epithelial to Mesenchymal Transition Regulated by ERK5 Signaling

**DOI:** 10.3390/biom11020183

**Published:** 2021-01-29

**Authors:** Akshita B. Bhatt, Saloni Patel, Margarite D. Matossian, Deniz A. Ucar, Lucio Miele, Matthew E. Burow, Patrick T. Flaherty, Jane E. Cavanaugh

**Affiliations:** 1Department of Pharmacology, School of Pharmacy, Duquesne University, Pittsburgh, PA 15282, USA; bhatta1@duq.edu; 2Department of Medicinal Chemistry, School of Pharmacy, Duquesne University, Pittsburgh, PA 15282, USA; patels4@duq.edu (S.P.); flahertyp@duq.edu (P.T.F.); 3Department of Medicine, Tulane University School of Medicine, New Orleans, LA 70112, USA; mmatossi@tulane.edu (M.D.M.); mburow@tulane.edu (M.E.B.); 4Department of Genetics and Stanley S. Scott Cancer Center, Louisiana State University Health Sciences Center, New Orleans, LA 70112, USA; ucardeniz@yahoo.com (D.A.U.); lmiele@lsuhsc.edu (L.M.)

**Keywords:** ERK5, EMT, cancer metastases, therapy

## Abstract

Extracellular signal-regulated kinase (ERK5) is an essential regulator of cancer progression, tumor relapse, and poor patient survival. Epithelial to mesenchymal transition (EMT) is a complex oncogenic process, which drives cell invasion, stemness, and metastases. Activators of ERK5, including mitogen-activated protein kinase 5 (MEK5), tumor necrosis factor α (TNF-α), and transforming growth factor-β (TGF-β), are known to induce EMT and metastases in breast, lung, colorectal, and other cancers. Several downstream targets of the ERK5 pathway, such as myocyte-specific enhancer factor 2c (MEF2C), activator protein-1 (AP-1), focal adhesion kinase (FAK), and c-Myc, play a critical role in the regulation of EMT transcription factors SNAIL, SLUG, and β-catenin. Moreover, ERK5 activation increases the release of extracellular matrix metalloproteinases (MMPs), facilitating breakdown of the extracellular matrix (ECM) and local tumor invasion. Targeting the ERK5 signaling pathway using small molecule inhibitors, microRNAs, and knockdown approaches decreases EMT, cell invasion, and metastases via several mechanisms. The focus of the current review is to highlight the mechanisms which are known to mediate cancer EMT via ERK5 signaling. Several therapeutic approaches that can be undertaken to target the ERK5 pathway and inhibit or reverse EMT and metastases are discussed.

## 1. Introduction

The epithelial to mesenchymal transition (EMT), one of the first steps in cancer metastases, is a continuum of morphologic transitions from a cobblestone-like epithelial state to a spindle-like mesenchymal state. The complex process of metastases involves EMT, intravasation in the blood vessels, survival in blood stream, extravasation at the secondary site, mesenchymal to epithelial transition (MET), and secondary tumor growth. Cancer metastases require high cellular plasticity and adaptability to survive in diverse physiological environments. It is important to identify targeted therapy to target different stages of metastases. Upregulation of EMT transcription factors via growth factors, epigenetic plasticity, and downregulation of tumor suppressor microRNAs (miRs) are a few mechanisms that drive EMT in cancer [1,2].

Epithelial and mesenchymal cells are characterized by differences in molecular markers expressed by these cells. E-cadherin and cytokeratins are markers of epithelial phenotype, whereas mesenchymal cells express N-cadherin, SNAIL, SLUG, and Vimentin among others. These markers can be co-expressed during cancer progression, leading to an intermediate epithelial/mesenchymal state, which is associated with poor prognosis [3]. EMT is regulated in several developmental and fibrotic processes during embryogenesis (Type-I EMT) and wound healing (Type-II EMT), respectively [4]. These processes are regulated by different context-dependent signals, for example, transforming growth factor (TGF)-β [5]. However, the discussion of these studies is beyond the scope of the current review and the focus of the current review is to discuss EMT in cancer (Type-III EMT). Upregulation of mesenchymal markers can occur due to mutations in components of the KRAS, Wnt, or EGFR signaling pathways. The suppression of E-cadherin is a central event during EMT. Epidermal growth factor (EGF) can mediate transcriptional activation of SNAIL via p21-activated kinase (PAK) [6]. TGF-β and RAS can co-operate to induce EMT in breast cancer [7]. Wnt/β-catenin, Jagged (JAG), and sonic hedgehog (SHH) signaling pathways are some examples of ligand-mediated signaling events, which can activate EMT in cancer via activation of downstream targets T-cell factor/lymphoid enhancer factor (TCF/LEF), NOTCH, or Gli-1 [8]. Several mechanisms that regulate EMT in cancer are reviewed in Figure 1. 

Tumor heterogeneity in terms of difference in driver mutations within the same cancer subtype and complexity of the tumor microenvironment has made the application of oncology therapeutics extremely challenging. Extracellular signaling factors and epigenetic effectors cooperate to initiate the EMT program and ultimately lead to metastases. One of the cellular adaptations during EMT involves increased capabilities of cancer cells to preferentially interact with the extracellular matrix, rather than the adjacent epithelial and stromal cells. Integrins are known to activate Src and focal adhesion kinases (FAK), which results in an increase in secretion of matrix metalloproteinases, loss of E-cadherin, and disruption of the adherens junctions (AJs) [9]. 

Tumor-associated chronic inflammation could initiate EMT via crosstalk between the inflammatory and tumor cells [10]. The tumor is infiltrated by diverse inflammatory and immune mediators. For example, 50% of the tumor is infiltrated with inflammatory macrophages. These infiltrated activated macrophages and proinflammatory T-cells can release cytokines, including transforming growth factor-β (TGF-β), tumor necrosis factor α (TNF-α), and interleukin-6 (IL-6), which are potent EMT inducers [11]. Expression of immune checkpoint proteins such as PD-L1 mediates the escape of cancer cells from natural killer (NK) cell-mediated cytotoxicity. In the given feedforward mechanism, EMT increases in PD-L1 expression via the microRNA-200-ZEB1 axis, which results in immune suppression and metastases [12].

Another feature that the cancer cells adopt as they transition to a mesenchymal state and enter the blood circulation for metastases is their ability to activate and bind platelets. At this stage, the cancer cells are termed as circulating tumor cells. Many studies argue that the cells are in an intermediate epithelial/mesenchymal state at this stage. There are coagulation-dependent mechanisms modulated via fibrin, which help the cancer cells to bind platelets and gain protection against the loss of anchorage-triggering anoikis, immune attack, and shear stress [13]. Drug resistant stem cells at the primary and metastatic sites are also known to possess EMT characteristics [14]. 

MicroRNA (miRNA) dysregulation is an important event during tumor progression. MicroRNAs can be classified as tumor suppressor or oncogenic. Loss of tumor suppressor and overexpression of oncogenic microRNAs are important events in regulating EMT in cancer. miRs-143 and -200 are frequently downregulated in several cancers. A possible mechanism for miR-143 downregulation includes the activation of KRAS and its downstream target Ras Responsive Element Binding Protein 1 (RREB1) [15]. Has-miR-200-3p is one of the fundamental EMT regulators and is known to target mesenchymal transcription factor ZEB1 [16].

## 2. Importance of ERK5 in Regulating Tumorigenesis

### 2.1. MEK5-ERK5 Pathway in Cancer

Extracellular signal-regulated kinase (ERK) 5, the newest member of the mitogen-activated protein kinase family (MAPK), is a marker for poor prognosis in cancer patients [17]. ERK5 is one of the members of the four MAPK signaling cascades, including ERK1 and ERK2, c-JUN N-terminal kinase (JNK) 1, and p38-MAPK. These cascades are regulated by three to five tiers of phosphorylation events, which are initiated by receptor tyrosine kinases and subsequent components of the MAPK family. While these pathways are under strict regulation by feedforward activation and feedback inhibition loops in healthy cells, these do not operate similarly in cancer cells. 

Although the ERK1/2 and ERK5 pathways share greater than 50% sequence homology at the N-terminal domain, ERK1/2 and ERK5 have been shown to mediate differential responses to growth factors, hypoxia, and pharmacological targeting with rapidly accelerated fibrosarcoma (BRAF) and MEK1/2 inhibitors. The ERK1/2 and ERK5 pathways regulate hypoxia-related genes via distinct mechanisms in normoxic versus hypoxic conditions [18]. These differences could be attributed to the unique C-terminal domain of ERK5, which contains two proline-rich regions and a nuclear localization sequence and facilitates transcriptional activation of oncogenes [19]. 

High ERK5 expression correlated with EMT, drug resistance, and poor patient survival in several cancers [20,21,22,23]. Whole-genome microarray analysis of patient samples revealed that overexpression of cyclin-dependent kinase 5 (CDK5), an important prognostic marker for the development of malignant colorectal cancer (CRC), could directly activate ERK5 and promote progression of colorectal cancer via activator protein-1 (AP-1) [24]. ERK5 negatively correlated with miR-143 expression and regulated proliferation, migration, and invasion in osteosarcoma [25,26]. DNA damage initiates the apoptotic signaling cascade via activation of ataxia-telangiectasia mutated (ATM) kinase, which is a cell cycle regulator and mediates phosphorylation of DNA damage and repair marker H2AX. Loss-of-function mutation in ATM is one of the causes of cancer, and ERK5 deletion in ATM-/- mice has been shown to delay tumorigenesis and increase response to DNA-targeting agents via H2AX phosphorylation in thymic lymphoma [27]. This was one of the few studies to identify the role of ERK5 in relation to tumor suppressors. 

### 2.2. Mechanisms for Dysregulated ERK5 Signaling in Cancer

Activating mutations in genes such as rapidly accelerated fibrosarcoma (RAF) (70% melanoma, 59% thyroid, 10% colon, and 6.7% lung cancer) [28] and rat sarcoma (RAS) (90% pancreatic, 50% thyroid, 30% lung, 15% ovarian, breast, liver, kidney cancer, and leukemias) have a major influence on MAPK signaling in several cancers. Most of these cancers have FDA-approved therapies as a line of treatment, but it becomes really challenging to treat these cancers once they metastasize. BRAF mutations in particular are known to mediate the EMT and metastases via hyperactivation of the MAPK, nuclear factor kappa B (NF-ᴋB), and phosphatidyl inositol-3-kinase (PI3K)/AKT pathways [29]. The ERK5 and AKT pathways are known to transactivate each other and regulate cell survival via phosphorylation and cytosolic sequestration of the pro-apoptotic protein Bad in triple negative breast cancer (TNBC) [30]. Moreover, the PI3K/AKT pathway is also known to mediate MEK5-ERK5 activation in malignant mesothelioma and neuroblastoma [31,32]. Hepatocyte growth factor (HGF) and its receptor c-met are also known to activate ERK5 and upregulate one of the downstream targets fos-related antigen-1 (FRA-1) via PI3K-AKT signaling in malignant mesotheliomas (MMs) [31]. HGF also induced ERK5 and protein tyrosine kinase (PTK) 6 activation and cell migration in breast cancer cells [33]. Platelet-derived growth factor (PDGF) BB increased bone morphogenetic protein (BMP) signaling in fibroblasts via PI3K-MEK1/2-MEK5-ERK5 activation [34]. Interleukin-6 is known to promote ERK5 activation and proliferation in multiple myeloma [35].

Cross-talk among the different members of the MAPK family, including ERK1/2 and ERK5, has been noted. ERK1/2 and ERK5 are known to commonly regulate downstream targets such as RSK, c-Fos, and CD-1. We have shown that novel inhibitors of the MEK1/2 and MEK5 pathways reverse the mesenchymal phenotype of MDA-MB-231 TNBC cells to epithelial and increase E-cadherin protein expression [36]. Some studies revealed that targeting the ERK1/2 pathway can lead to a compensatory increase in ERK5 activation via upregulation of c-MYC or insulin growth factor receptor (IGFR) [37,38,39,40]. ERK5 can compensate for targeting ERK1/2 to inhibit upregulation of macrophage colony stimulating factor receptor (M-CSFR)-mediated macrophage differentiation in acute myeloid leukemia cells [1,4]. Inhibition of both ERK1/2 and ERK5 has been found to be necessary for efficient targeting of neuroblastoma rat sarcoma (NRAS) and BRAF-mutant melanomas [37,40,41,42]. On the contrary, ERK5 has been found to be downstream of the BRAF-MEK1/2-ERK1/2 signaling in melanoma and ERK1/2 is known to promote nuclear localization of ERK5 in HEK293 and PC12 cells via Thr732 phosphorylation [43,44]. Although the MEK1/2-ERK1/2 pathway is fairly understood in regulating EMT in cancer, the involvement of the MEK5-ERK5 pathway in EMT is overlooked (293 versus 22 results in Pubmed search). 

Hyperactivation of ERK5 could be a result of downregulation of tumor suppressor proteins, microRNAs, or phosphatases/proteases that inhibit ERK5 via a negative feedback loop. MicroRNAs (miRs) are a class of short non-protein-coding single-stranded RNAs and their function is to silence protein expression post-transcriptionally. These bind with the 3’-untranslated region (UTR) of the target mRNAs, leading to their translational repression or destabilization. Loss of tumor suppressor miRs is an important mechanism responsible for the overexpression of oncoproteins. Some studies suggest that ERK5 is negatively regulated by tumor suppressor miRNAs miR-143 and miR-200 in breast cancer and glioblastoma, respectively [45,46]. 

Downregulation of transcription factor special AT-rich sequence-binding protein 2 (SATB2) via miR-31, miR-34, or TGF-β signaling has been indicated to be associated with cancer progression [47]. High SATB2 expression correlates with favorable prognosis and enhanced chemosensitivity in colorectal cancer (CRC) [47,48]. Overexpression of SATB2 increased the epithelial marker E-cadherin, decreased mesenchymal markers vimentin and N-cadherin in CRC cell lines in vitro, and suppressed metastases in vivo [49]. SATB2 was identified to inhibit ERK5 activity and decrease CRC cell migration, invasion, and colony formation in vitro and tumor progression in vivo [50]. However, the total ERK5 expression was unaffected, indicating that SATB2 may not transcriptionally regulate ERK5. Finally, ERK5 degradation is mediated by the tumor suppressor von Hippel-Lindau (VHL) through prolyl hydroxylation-dependent ubiquitination and subsequent proteasomal degradation [51]. The VHL gene is often mutated in clear cell renal cell carcinoma (CCRCC). Hypoxia-inducible factor-1α (HIF-1α), one of the major targets of VHL-mediated degradation, mediates the survival of cancer cells under low oxygen conditions [52]. Disease-specific survival was greater with high ERK5 and HIF-1α expression and low VHL expression in CCRCC patients, indicating ERK5 as an essential target in CCRCC. 

## 3. Mechanisms of EMT Regulated by ERK5

The role of MEK5/ERK5 signaling in tumor progression is relatively well-understood; however, its role in EMT and tumor metastasis is only recently explored. ERK5 is a big MAPK with diverse cellular functions including cell cycle progression, proliferation, and tumor progression. TNFα-resistance and estrogen receptor suppression were identified as key mechanisms to induce MEK5-ERK5-mediated EMT phenotype in breast cancer [53,54]. Recently, clustered regularly interspaced short palindromic repeats (CRISPR) and kinome-wide high-content short interfering RNA (siRNA) screens have identified ERK5 as a critical prognostic factor for EMT and metastases in KRAS mutant non-small cell lung cancer (NSCLC) and breast cancer, respectively [55]. In a translational study, genes involved in the MAPK pathway, TGF-β signaling, and signaling pathways involved in apoptosis were found to be significantly upregulated in oral squamous cell carcinoma (OSCC) compared to healthy tissue. In the same study, high ERK5 but not high ERK1 expression was positively associated with advanced tumor stage and lymph node metastases. 

Signal transducer and activator of transcription (STAT) 3, which binds directly to the promoter region of MEK5, is frequently upregulated in breast cancer tissues compared to the healthy controls [56]. STAT3 upregulation decreased the epithelial marker E-cadherin and increased mesenchymal markers vimentin, SNAIL, SLUG, and ZEB and cell migration and invasion via the MEKK2/3-MEK5-ERK5 signaling axis in breast cancer cells [57]. ERK5 regulates EMT and the generation of circulating tumor cells in lung and breast cancer, which allows the cancer cells to survive in the blood during metastasis and gain resistance against fluid shear stress [58,59].

Dual specificity phosphatases (DUSPs) are a class of enzymes that participate in dephosphorylating the target kinase. DUSP6 downregulation is associated with poor prognosis and worse overall survival in NSCLC patients. It has recently been shown that ERK5 was an essential target of DUSP6 in the H460 NSCLC cell line. Downregulation of DUSP6 enhanced ERK5 activation and EMT, and forced expression of DUSP6 decreased ERK5 activation and EMT characteristics of NSCLC cells, accompanied with respective changes in cell morphology, migration, and adhesion to the extracellular matrix (ECM) [60].

## 4. ERK5 in Regulating the Metastatic Cascade and Therapeutic Interventions

The role of ERK5 in regulating the EMT axis is only recently being explored. Increasing evidence in the last decade suggests that ERK5 expression and activation have an essential role in regulating tumor characteristics that promote EMT and metastases, including morphological shift, degradation of the extracellular matrix, stemness and anchorage independent growth, angiogenesis, and immune evasion. The mechanisms that are responsible for the activation of ERK5 and subsequent EMT are outlined in Figure 2. All these events increase cell migration, invasion, and metastasis.

### 4.1. ERK5 Signaling in Cytoskeletal Rearrangement

ERK5 was found to associate with actin and cofilin in the cytosol in TNBC breast cancer cells, whereas ERK5 interacted with estrogen receptor (ER)-α in the nucleus in ER-positive breast cancer to increase cell invasion [61]. This study characterized the role of ERK5 in regulating EMT based on ER status; however, the TNBC cell lines that were utilized have an epithelial phenotype. Subcellular location-specific roles of ERK5 with respect to phenotypic characteristics require further research. TGF-β has been studied as a potent EMT inducer in various contexts in several cancer models [5,62,63]. The MEK5-ERK5 signaling cascade was identified as an essential effector of TGFβ-mediated EMT, using a high-content microscopy screen and a kinome-wide siRNA library, in normal mammary epithelial cells and breast cancer cells [64]. Targeting the MEK5-ERK5-MEF2B pathway decreased TGF-β-mediated induction of EMT, as indicated by a phenotypic switch from epithelial to mesenchymal, cell migration, and lung metastases in breast cancer [64]. TGF-β1, via anaplastic lymphoma kinase (ALK) 5 receptor and p38, has been shown to increase the association of MEK5 with ERK5 and its downstream substrate monocyte enhancer factor-2C (MEF2c) in human primary renal proximal tubule epithelial cells (PTECs) and HKC-8 transformed cells [65].

Metacore GeneGo enrichment analysis identified that ERK5 positively regulated the proteins essential for actin cytoskeleton regulation, gelsolin, N-WASP, p-PLK1, and SPA1, and promoted a phenotypic switch from epithelial to mesenchymal in lung cancer cells [66]. MEK5 and ERK5 activation was also identified to upregulate TGF-β-mediated induction of EMT in a feedforward loop in lung cancer cells [66]. ERK5 activated focal adhesion kinase (FAK) at S910 position and upregulated upstream stimulatory factor (USF)1-mediated migration and invasion in metastatic lung cancer cells [66]. Transfection with dominant negative MEK5 and ERK5 isoforms simultaneously led to a significant reduction in lymph node and lung metastases of melanoma xenografts [66].

Proto-oncogene tyrosine protein kinase sarcoma (SRC) protein is an important regulator of the actin cytoskeleton [67]. Rho inactivation by Rho Guanosine-5’-triphosphatases (GTPases) and Rho GTPase activating proteins (GAPs) is responsible for an increase in stress fibers, morphological transition, and cell invasion. Rho-GTPases act as activators or inhibitors of biochemical pathways depending upon the extracellular signal. Activation of SRC via ectopic expression or mutation, as seen in many cancers, can suppress Rho activation and increase the formation of podosomes, which are F-actin rich structures on the cells, and cell invasion [68,69] Activation of the ERK5-MEF2c axis, but not ERK1/2, was mainly responsible for in v-SRC-transformed NIH-3T3 cells via induction of deleted in liver cancer 1 (DLC-1) expression, a negative regulator of Rho [69]. A similar observation has been reported in fibroblasts, explaining the role of ERK5 in regulating the actin cytoskeleton [70]. Invadopodia formation on invasive tumor cells allows for ECM degradation. ERK5 overexpression increased invadopodia formation and metastases of PC-3 prostate cancer xenografts to lymph nodes and lungs [71]. Moreover, ERK5 downregulation in human melanoma cells was found to decrease invadopodia formation and gelatin degradation [71].

### 4.2. ERK5 Signaling in Mediating Adhesion to the ECM

CRISPR/Cas9-mediated knockout of ERK5 resulted in a decrease in the gene transcription and mRNA synthesis of matrix-associated proteins, including collagenases, matrix metalloproteinases, and EMT markers in TNBC [72]. There was a decrease in the formation of tumors due to the lack of ECM and complete loss of metastatic dissemination to lungs and livers in an MDA-MB-231 xenograft model in vivo [72]. The MAPK-ERK5 pathway is associated with upregulation of fos-related antigen (FRA)-1, which is known to be overexpressed in most aggressive forms of breast cancer, including TNBC [73,74,75]. FRA-1 increased cell protrusion to facilitate tumor cell invasion through the collagenous matrix by decreasing the RhoA-ROCK pathway and β1-integrin activity via ERK [73].

Collagens act through β1-integrin/FAK/Src signaling to suppress epithelial characteristics, including cadherin-mediated cell–cell adhesion, and increase cell motility [76]. β1-integrin signaling via the vitronectin (VN) receptors increased FAK and ERK5 activation and promoted TNBC and prostate cancer cell adhesion to the ECM and micromotion in vitro [77] Adhesion of cancer cells to ECM was inhibited in the presence of VN blocking antibodies [77]. Moreover, there was a reduction in attachment of TNBC and prostate cancer cells to vitronectin and fibronectin in ERK5 knockout cells via reduction in the FAK/proline-rich tyrosine kinase 2 (PYK2) interaction [78]. Metastatic ovarian cancer is regulated by overexpression of human epidermal growth factor receptor (HER2). The silencing of HER2 in SKOV-3 ovarian cancer cells led to a significant increase in PTPN12 phosphatase, as identified by DNA microRNA targeting all the tyrosine phosphatases (PTPs) and dual-specificity phosphatases (DUSPs) of the human genome. PTPN12 regulated FAK phosphorylation and decreased cell migration in vitro. Further analysis revealed the role of ERK5 in mediating FAK phosphorylation in a HER2-dependent manner [79].

MEK5 overexpression was associated with poor patient survival in prostate cancer. MEK5 was weakly expressed in benign glandular epithelial cells, whereas it was overexpressed in malignant glandular epithelium. Moreover, higher MEK5 overexpression was correlated with an increase in bone metastases of prostate cancer [21]. ERK5 is overexpressed in osteosarcoma samples compared to the healthy controls [25]. Matrix metalloprotease (MMP) upregulation is an important event for the degradation of extracellular matrix. ERK5 knockdown was found to decrease MMP-9 mRNA and protein expression, cell invasion, and lung metastases in osteosarcoma. These effects were mediated via SLUG downregulation and there was no effect on E-cadherin protein expression [80]. 

### 4.3. ERK5 Signaling in Regulating Stemness 

MEK5 and ERK5 were overexpressed in colon cancer tissues compared to healthy controls; ERK5 expression specifically correlated with an increased metastatic and invasive potential in colon cancer patients [81]. Constitutive activation of ERK5 resulted in an increase in the nuclear accumulation of NF-kB-mediated upregulation of vimentin expression, and an increase in cell cycle progression and cell migration in colon cancer cells [81]. Residual stem cell populations in colon cancer patients have made the development of therapeutic targeting very challenging. Pharmacological targeting of MEK5/ERK5 signaling via the ERK5 inhibitor XMD8-92 has been reported to decrease the expression of SOX2, NANOG, OCT4, and ALDH, the markers of pluripotency, via inhibition of the NF-kB/Interleukin-8 (IL-8) signaling axis [82]. Inhibition of the ERK5 pathway functionally corresponded to a decrease in tumorsphere formation and an increase in 5-fluorouracil chemosensitivity in aggressive colon cancer cell lines [82]. 

Loss of tumor suppressor SATB2 is correlated with increased metastatic potential of colon cancer [48]. Ectopic expression of SATB2 in colon cancer cells resulted in a decrease in ERK5 activation, colony formation, migration, and invasion in colon cancer cells and primary tumor growth in colon cancer xenografts in vivo [50]. Protein tyrosine kinase (PTK) 6 expression positively correlated with E-cadherin expression in colon cancer patients [83]. PTK6 knockdown resulted in a decrease in E-cadherin and ZO-1 epithelial markers and an increase in vimentin, ZEB1, and claudin-1 mesenchymal markers via STAT3-MEK5/ERK5 in colon cancer cells, which was reversed by PTK6 reintroduction. These effects were accompanied by an increase in cell migration, colon sphere formation in soft agar, and tumor mass in the shPTK6 group versus the control [83]. miR-200 negatively correlated with ERK5 expression in glioma patient tissues. miR-200 upregulation decreased in ERK5-mediated EMT and colony formation in glioblastoma cells in vitro [46].

ERK5 is upregulated in malignant mesothelioma tumor samples compared to healthy controls [20]. XMD8-92 decreased tumor sphere formation and invasion via doublecortin-like kinase 1 (DCLK1) downregulation in malignant mesothelioma cell lines in vitro. Pancreatic ductal adenocarcinoma (PDAC), one of the most aggressive and chemo-resistant cancers, is in dire need of targeted therapy. DCLK1 is upregulated in PDAC; XMD8-92 was found to decrease EMT transcription factors ZEB1/2, SNAIL, and SLUG via DCLK1 and c-Myc suppression in highly aggressive AsPC-1 metastatic PDAC cells. Moreover, XMD8-92 inhibited stemness regulators OCT4, SOX2, NANOG, and NOTCH and angiogenesis mediators vascular endothelial growth factor receptors (VEGFR)1/2 in an AsPC-1 PDAC xenograft model in vivo [84]. ERK5 inhibition also decreased stem cell marker expression in bone marrow mononuclear cells (BMMCs) derived from chronic myeloid leukemia (CML) patients [85]. 

### 4.4. Role of ERK5 in Regulating the Tumor Microenvironment 

ERK5 is overexpressed in CD163+ tumor associated macrophages in bladder, lung, and breast cancer patient samples [86]. ERK5 increased 12-O-tetradecanoylphorbol-13-acetate (TPA)-induced chemokine (C-X-C motif) ligand (CXCL)1, CXCL2, IL-1α, IL-1β, and COX-2 transcription in neoplastic epidermis, indicating its role in regulating the tumor immune microenvironment [87]. ERK5 promoted the transition of anti-tumorigenic M1 macrophages to a pro-tumorigenic M2 phenotype, which was reversed by ERK5 deletion via inhibition of STAT3 at Tyr705 position in TAM-cancer cell co-culture models. Moreover, the deletion of ERK5 increased T-cell infiltration via upregulation of CCL5 and CXCL10 chemokines in a prostate cancer model [88].

Colony-stimulating factor (CSF) 1 was studied to induce ERK5-Src-mediated proliferation in macrophage cells of mesenchymal lineage with high CSFR-1 expression. Granulocyte-macrophage (GM)-CSF is known to activate MAPK-ERK signaling and EMT in colon cancer; [89] however, the involvement of ERK5 in CSF-mediated EMT in cancer warrants further investigation. Asbestos can activate a nod-like receptor family member containing a pyrin domain 3 (NLRP3) inflammasome in lung cancer, which increases the secretion of inflammatory mediators including IL-1β. Malignant mesotheliomas are characterized by an increase in inflammatory mediators and the ERK5 signaling pathway. Targeting the ERK5 pathway with XMD8-92 resulted in a decrease in asbestos-induced IL-1β secretion, colony formation, and tumor growth in malignant mesothelioma [90]. TGF-β can signal in a paracrine manner to promote invasion via AP-2a and ERK5 in cholangiocarcinoma [91].

α-smooth muscle actin (α-SMA), fibroblast associated protein (FAP)-positive cancer associated fibroblasts (CAFs), and M2 macrophages together exacerbate cancer prognosis and metastases in several cancers [92]. Secreted growth factors from fibroblasts, including fibroblast growth factor, can promote MMP secretion and angiogenesis via the MEK/ERK signaling pathway [93]. An increase in CAFs resulted in an increase in the number of infiltrating macrophages and CD8-positive T cells, consistent with an increase in proinflammatory cytokine levels of IL-1α, IL-1β, IL-6 and tumor necrosis factor (TNF) in colon cancer [93]. Moreover, CAFs were recently identified to regulate PD-L1 expression in colorectal cancer cells via ERK5 activation [94]. These effects together can have a greater implication in driving cancer metastases. Vascular growth factor is known to be upstream of ERK5 and regulates tumor angiogenesis. Fibroblasts and endothelial cells secrete TSP1, an anti-angiogenic factor, which is inhibited by ERK5, indicating the role of ERK5 in mediating angiogenesis [95]. ERK5 knockout in triple negative breast cancer tumor xenografts resulted in a decrease in CD36, an angiogenesis marker. Several mechanisms of ERK5-mediated EMT are summarized in Table 1.

## 5. Conclusions

In the present review article, we have highlighted the role of the newest MAPK member ERK5 in various stages of the metastatic cascade: initiating with EMT, immune escape, increased production of collagen degrading enzymes, MMPs, collagenases, form and structure of the ECM, angiogenesis, survival in blood circulation, and secondary tumor growth at metastatic sites. Pharmacological inhibitors of the MEK5-ERK5 pathway SC-1-181, SC-1-151, XMD8-92, JWG-045, BIX02189, microRNAs -143, -200, and natural products resveratrol, curcumin, and vitamin D are a few candidates which can be used to target ERK5-mediated EMT in cancer research. Pharmacological interventions known to inhibit or reverse EMT and metastases are summarized in Table 2. 

In summary, SC-1-151, a dual MEK1/2 and MEK5 inhibitor, was found to reverse the mesenchymal phenotype, decrease mesenchymal markers, and increase epithelial marker E-cadherin in triple-negative and tamoxifen-resistant breast cancer [36]. XMD8-92, an ERK5 inhibitor, decreased stem-cell properties in colon cancer, inhibited EMT in PDAC, and attenuated inflammasome formation in malignant mesothelioma via downregulation of oncogenes c-MYC, KRAS, and VEGFR1/2, stemness regulators NOTCH1, OCT4, SOX2, and NANOG, EMT transcription factors ZEB1/2, SNAIL, and SLUG, and upregulation of tumor suppressor miRNAs -let-7a, -143/145, and -200 [20,84]. BIX02189, a MEK5 inhibitor, inhibited TGF-β induced EMT in breast cancer. Several natural products have been identified to attenuate EMT induced by benzidine. Resveratrol decreased smoke induced EMT in bladder cancer via STAT3/Twist1 inhibition [96]. Curcumin inhibited smoke-induced EMT as shown by an increase in the epithelial marker E-cadherin and zona occludens-1 (ZO-1) and a decrease in mesenchymal markers vimentin and N-cadherin in bladder cancer [97,98,99]. Vitamin D promoted the macrophagic differentiation of monocytes in acute myeloid leukemia (AML) via ERK5-mediated CSFR upregulation, thus promoting cancer cell phagocytosis. microRNA-143 or -200 restoration reversed EMT in osteosarcoma or glioblastoma, respectively, via ERK5 downregulation. These studies have demonstrated the relevance of targeting ERK5 to prevent or reverse EMT and metastases, thus validating ERK5 as an important therapeutic target in cancer. 

Some of the ERK5 inhibitors were recently found to promote the nuclear translocation of ERK5 [100]. Hence, there is still a need for the discovery of effective ERK5 inhibitors/degraders. While these studies have been extremely helpful in identifying the less well-understood effects of ERK5 signaling via several mechanisms, there are several avenues that deserve more research in EMT and drug development; these include the cytosolic versus nuclear functions of ERK5 with respect to EMT, the role of total ERK5 versus phosphorylated ERK5, and its crosstalk with the immune system. The studies reviewed here show that ERK5 is a promising target in regulating EMT and metastases, and there is an urgent need to develop clinical drug candidates for targeting the ERK5 pathway.

## Figures and Tables

**Figure 1 biomolecules-11-00183-f001:**
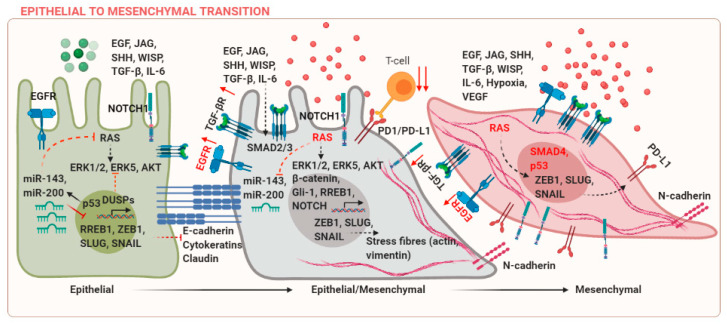
Mechanisms regulating epithelial to mesenchymal transition (EMT) in cancer. Created with BioRender.com.

**Figure 2 biomolecules-11-00183-f002:**
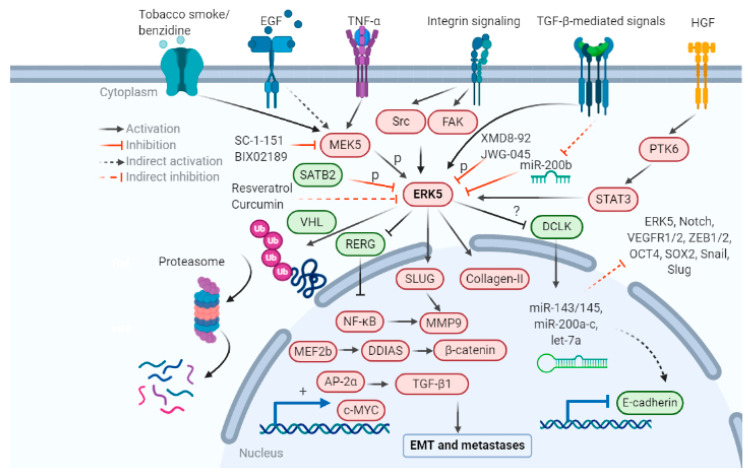
Upstream and downstream regulators of ERK5-mediated EMT. ERK5 activation is regulated by growth factors, integrins, and carcinogens such as tobacco smoke. Activation or overexpression of ERK5 can regulate gene expression of several mesenchymal genes, leading to suppression of E-cadherin and subsequent activation of the EMT program in cancer. Inhibitors of the MEK5-ERK5 cascade known to inhibit or reverse EMT are also included in the figure. Created with BioRender.com.

**Table 1 biomolecules-11-00183-t001:** Mechanisms modulating the extracellular signal-regulated kinase (ERK5) pathway and effect of targeting ERK5 on EMT, and metastases.

Model	Upstream Regulator(s) of ERK5	Intervention	Targets	EMT	Viability/Proliferation/Apoptosis	Ref.
**Breast cancer**						
MDA-MB-231, Hs-578T tumor xenografts	ER-, PR-, HER2-	CRISPR ERK5 knockout	COL1A1, COL4A1, and ITGA1	Morphology, migration, ECM formation, and CD31	No effect in vitro, decreased ECM formation and primary tumor growth and lung and liver metastases	[72]
MDA-MB-231 cells	ER-, PR-, HER2-, BRAF, RAS,	VN receptor blocking antibodies	VN, FAK, ERK5	Adhesion and haptotactic micromotion	Unknown	[77]
MCF-10A, nMuMG/E9, MDA-MB-231 cells, PymT-1099 cells in vitro and in vivo 4T1 mouse model	TGF-β	siERK5, siMEK5, BIX02189 (MEK5 inhibitor)	TGF-β, MEK5, ERK5, MEF2B	Morphology, migration, and lung metastases	Primary tumor growth unaffected	[64]
MDA-MB-345, MDA-MB-231 cells	HGF	siERK5	HGF, PTK6, ERK5	Migration	Unknown	[31]
MCF-7, BT-474, T47D, MDA-MB-468, MDA-MB-345, SKBr3	Estrogen	siERK5, XMD8-92	TFF1, PgR, GREB1, RET, PMAIP1, cofilin, F-actin	Migration, invasion, and colony formation	Cell cycle	[61]
MDA-MB-231, BT-549, TAMR-MCF-7	ER-, PR-, HER2-	Compound 1 (SC-1-151)—dual MEK1/2 and MEK5 inhibitor	ZEB1, SNAIL, Vimentin	Migration, spheroid formation, colony formation	Cell viability and proliferation	[36]
**Glioblastoma**						
Patient samples, U87 and U251 cells	ERK5	miR-200	ERK5	E-cadherin, N-cadherin, vimentin	Decrease in cell invasion and colony formation	[46]
**Colon cancer**						
Patient tissue, SW-480 cells, xenograft	PTK6	shPTK6	pERK5, STAT3, ZEB1, claudin-1, vimentin	Morphology, migration, tumorsphere formation	Ki67-mediated proliferation and primary tumor mass	[83]
HCT-116, DLD-1 cells	SATB2	SATB2 OE	pERK5	Morphology and colony formation	Primary tumor growth	[50]
HCT116, HT29, SW480, and SW620 cells		XMD8-92 (ERK5 inhibitor)	NF-ᴋB, IL-8, SOX2, OCT4, NANOG	Tumorsphere formation	Apoptosis in anchorage-independent condition	[82]
HT29, HCT-116		XMD8-92, PD-L1	ERK5, α-SMA	Colony formation	Increased apoptosis and decreased viability and proliferation	[94]
**Ovarian cancer**						
SKOV-3 cells	HER2	shHER2, sh ERK5	PTPN12, pFAKS910	Migration	Unknown	[79]
**Pancreatic cancer**						
AsPC-1 cells and tumor xenografts in vivo	KRAS, p53	XMD8-92	c-Myc, DCLK1, SOX2, OCT4, NANOG, NOTCH, VEGFR1/2	Decrease in cell migtation and invation	Decrease in primary tumor growth	[84]
**Prostate cancer**						
LNCaP cells and patient samples	PTEN	MEK5 OE	AP-1, MMP9	Increase in cell migration, invasion	Increase in cell proliferation	[21]
Primary and metastatic tumor samples, PC-3 xenograft	Nuclear ERK5	ERK5 OE, miR-143	MMP9	Invadopodia formation, ECM degradation, invasion, lung and lymph node metastases	Unknown	[71]
**Osteosarcoma**						
U2OS cells		ERK5 KD and OE	ERK5, SLUG, MMP-9	Migration, invasion, and lung metastases	No change	[80]
SOSP-M cells		ERK5 KD and OE, miR-143	ERK5	Migration and invasion	Proliferation	[25,26]
**Cholangiocarcinoma**						
Patient samples, HCCC, and RBE tumor xenografts	TGF-β	miR-200	TGF-β, ERK5, α-SMA, fibronectin, TTF-1, and E-cadherin	Migration and invasion		[91]
**Leukemia**						
KCL22, K562, and LAMA-84 cells		XMD8-92, JWG-045, BIX02189 + imatinib (TKI), shERK5	p21, p27, CD34+ BMMCs	Decrease in colony formation	Decrease in cell viability	[85]
Patient samples, HL60, and U937 cells		Vitamin D, XMD8-92, BIX02189, UO-126, PD98059	M-CSFR	Monocyte to macrophage differentiation	Increased phagocytosis	[41]
**Renal cell carcinoma**						
Patient samples, 786-0, 769-P, and Caki-2 cells	VHL	siERK5	HIF-1α	Migration	Unknown	[51]
**Lung cancer**						
Patient samples, A549 and H1299 cells, xenografts		DNERK5 and MEK5A	pERK5/FAK/ USP, gelsolin, N-WASP, p-PLK1, and SPA1	F-actin polymerization, migration, invasion, and metastases	Unknown	[66]
H460 NSCLC cells	KRAS, EGFR	shDUSP6	pERK5, VAV3, SNAIL, SLUG, ZEB1	Cytoskeletal rearrangement, adhesion, migration, and invasion	Decrease in tumor volume/ metastases unknown	[60]
Patient tissues, H290 and H2052 cells	MET, ERK5, DCLK	XMD8-92, XL184 (MET inhibitor)	DCLK1	Invasion, tumor sphere	Decrease in cell viability	[20]
Patient samples, HP-1, H2373, H2461, and H2595 cells	Inflammation	XMD8-92	IL-6, IL-1β, FGF2, TFPI2	Colony formation	Tumor growth	[87]

**Table 2 biomolecules-11-00183-t002:** Pharmacological interventions for targeting the ERK5 pathway to inhibit or reverse EMT.

Pharmacological Intervention	Cancer	Target	Effect on EMT/MET
SC-1-151	Triple-negative and tamoxifen-resistant breast cancer	MEK1/2 and MEK5	EMT reversal
JWG-045	Leukemia	ERK5	Inhibition of EMT
XMD8-92	Colon cancer, pancreatic cancer, breast cancer, leukemia, and lung cancer	ERK5 and BRD4	Inhibition of EMT, stemness, and metastases
BIX02189	Breast cancer, leukemia	MEK5	Inhibition of TGFβ induced EMT
Resveratrol	Bladder cancer	STAT3/Twist-1	Reversal of smoke induced EMT
Vitamin D	Acute myeloid leukemia	M-CSFR	Macrophage differentiation
Curcumin	Lung cancer, bladder cancer	ERK5/AP-1, ERK1/2, c-Fos, c-Jun, p-38	Inhibition of smoke induced EMT
miR-143	AML, Cholangiocarcinoma, and osteosarcoma	ERK5	Inhibition of differentiation and apoptosis of AML cells; targets migration and invasion in osteosarcoma
miR-200	Glioblastoma	ERK5, ZEB1	EMT reversal

## Data Availability

Not applicable.

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
