# Peer review of "Molecular Mechanisms of Epithelial to Mesenchymal Transition Regulated by ERK5 Signaling"

_biomolecules, 2021, doi:10.3390/biom11020183_

Round 1

Reviewer 1 Report

This is a review article on the mechanisms regulating epithelial mesenchymal transition particularly with respect to carcinogenesis. Investigations of specific mechanisms are an important topic of study and the potential for an epithelial tumor to transdifferentiate to a mesenchymal phenotype is a critical factor in tumor treatment and the response to specific therapies. There is an excellent review of other publications on the role of ERK5 in tumorigenesis and this information is linked to potential mechanisms that could be involved.  There are several potential linkages of the ERK5 pathway to tumor metastasis but the evidence is indirect and the conclusions stated are not based on original new research but on implications based on the results of many other investigators.  Linkages of the upregulation of several different genes and potential linkages to a mechanism for tumorigenesis are speculative and not supported by original research findings, while this is frequently seen in some review articles the potential mechanisms presented are primarily speculative rather than derived for new research findings.  It is important that it is clearly stated that these potential mechanism are speculative and based on the use of results from several other investigators to develop a potential molecular model.  The potential mechanisms that are presented and discussed would be very appropriate hypotheses to be tested in original research and it should be made clear that many of the conclusions presented are highly speculative. Multiple other mechanisms related to the ECM and cell phenotype are also presented but the linkage to the overall theme of the manuscript is not clearly defined and further emphasizes the speculative nature of the manuscript.  Many examples of altered ERK5 regulation in tumors are presented but these are again not evidence of a specific mechanism but rather evidence for genetic dysregulation that is commonly seen in tumors, it is much more important to provide the evidence that links the specific changes in gene expression with the mechanism of tumorigenesis.  The mechanism of EMT is important in many situations and particularly during embryogenesis, which is not mentioned in this manuscript but does demonstrate that EMT is a normal embryological event and in tumor situations that process is inappropriately expressed, this absence of perspective about the mechanisms of tumorigenesis and EMT is a significant weakness of this manuscript.

This is an interesting article that presents a potential mechanism that could be involved in tumorigenesis however the conclusions presented are primarily speculation/hypothesis and not supported by original research findings.

Author Response

Thank you for your review report. The potential mechanisms described in the text, figures, and table are solely retrieved from original papers and none of the statements are speculations. Appropriate references are provided after every mechanism is described. ERK5 is involved in regulating EMT by several mechanisms, which are reviewed in this paper. One place where we have a speculation is whether ERK5 directly or indirectly regulates the expression of DCLK and therefore, we have already placed a ‘?’ in the signaling figure to denote that ERK5 may possibly regulate DCLK levels. Regarding involvement of EMT in normal tissues, we have modified the text to include about Type-I and Type-II EMT; however, significant focus of the review is EMT in cancer (Type-III EMT). We have further expanded the conclusion to include the mechanisms by which ERK5 inhibitors prevented and reversed EMT in several cancer models.

Reviewer 2 Report

The authors of the review article describe the role of ERK5 signaling in epithelial to mesenchymal transition. The article comprehensively describes the ERK5 mechanisms of action, but sometimes it is difficult to follow. Several issues need to be explained.

  • In the introduction section the authors should better describe the molecular mechanisms of epithelial to mesenchymal transition. They should write about epithelial factors, mesenchymal factors and transcription factors involved in EMT. With that fragment molecular mechanisms of action of ERK5 will be easier to follow. New figure presenting those factors will be useful.
  • Short fragment explaining microRNAs role should be added in the introduction section.
  • The names of proteins – transcription factors involved in EMT in human should be written in capital letters, e.g. SNAIL, SLUG.
  • Figure legends should be better described.
  • Chapter 1.2 should be integrated with chapter 2 – they are both about ERK in cancer.
  • Gene abbreviations in the manuscript should be explained when first mentioned.
  • Reference 11 is autocitation of the manuscript that has been not published. It should be removed if the manuscript has not been already published.
  • Misspelling should be corrected (e.g. cancer metastases requires – line 39) and double spaces should be removed.
  • Graphical abstract for the article should be prepared.

Reviewer 3 Report

Brief summary

In the present review article, the authors give a complete overview of the role of ERK5 in various stages of the metastatic cascade, and the effect of targeting ERK5 on EMT and metastases. They highlight the different mechanisms for dysregulated ERK5 signaling in cancer and then focus on the role of ERK5 in the metastatic cascade: initiating with EMT, immune escape, increased production of collagen degrading enzymes MMPs, collagenases, form and structure of the extracellular matrix, angiogenesis, survival in blood circulation, and secondary tumor growth at metastatic sites. These effects are presented in a table summarizing these different mechanisms in different types of cancers, and two graphical illustrations that present simultaneously the upstream and downstream regulators of ERK5-mediated EMT signaling and the regulation of the tumor microenvironment via ERK5.

Broad comments

This is a well-written review that summarizes a large number of relevant studies highlighting the mechanisms which are known to mediate cancer EMT via ERK5 signaling. As mentioned before, the authors give a complete overview of the role of ERK5 in various stages of the metastatic cascade, and the effect of targeting ERK5 on EMT and metastases. These effects are presented in a comprehensive and easily read table and two graphical summaries that contribute to enhancing the quality of the manuscript.

In the conclusion, the authors cite some potential candidates, which can be used to target ERK5-mediated EMT in cancer research. It would enhance the clarity of the review if this part was further extended and developed.

Specific comments

In order to clarify the part regarding the several therapeutic approaches that can target the ERK5 pathway and inhibit or reverse EMT and metastases, it would be really helpful to resume these different approaches in one table.  
